# A Splice Variant of NCOR2, BQ323636.1, Confers Chemoresistance in Breast Cancer by Altering the Activity of NRF2

**DOI:** 10.3390/cancers12030533

**Published:** 2020-02-26

**Authors:** Man-Hong Leung, Ho Tsoi, Chun Gong, Ellen PS Man, Stefania Zona, Shang Yao, Eric W.-F. Lam, Ui-Soon Khoo

**Affiliations:** 1Department of Pathology, Li Ka Shing Faculty of Medicine, The University of Hong Kong, Hong Kong; george09@connect.hku.hk (M.-H.L.); frankie_tsoiho@yahoo.com.hk (H.T.); cg602@cam.ac.uk (C.G.);; 2Department of Surgery and Cancer, Imperial College London, Hammersmith Hospital Campus, London W12 0NN, UK

**Keywords:** splice variant BQ, chemoresistance, breast cancer, NRF2, NCOR2

## Abstract

Breast cancer is the most common type of female cancer. Reactive oxygen species (ROS) are vital in regulating signaling pathways that control cell survival and cell proliferation. Chemotherapeutic drugs such as anthracyclines induce cell death via ROS induction. Chemoresistance development is associated with adaptive response to oxidative stress. NRF2 is the main regulator of cytoprotective response to oxidative stress. NRF2 can enhance cell growth, antioxidant expression, and chemoresistance by providing growth advantage for malignant cells. Previously, we identified BQ323636.1 (BQ), a novel splice variant of nuclear co-repressor NCOR2, which can robustly predict tamoxifen resistance in primary breast cancer. In this study, we found that BQ was overexpressed in epirubicin-resistant cells and demonstrated that BQ overexpression could reduce the levels of epirubicin-induced ROS and confer epirubicin resistance. In vivo analysis using tissue microarray of primary breast cancer showed direct correlation between BQ expression and chemoresistance. In vitro experiments showed BQ could modulate NRF2 transcriptional activity and upregulate antioxidants. Luciferase reporter assays showed that although NCOR2 repressed the transcriptional activity of NRF2, the presence of BQ reduced this repressive activity. Co-immunoprecipitation confirmed that NCOR2 could bind to NRF2 and that this interaction was compromised by BQ overexpression, leading to increased transcriptional activity in NRF2. Our findings suggest BQ can regulate the NRF2 signaling pathway via interference with NCOR2 suppressive activity and reveals a novel role for BQ as a modulator of chemoresistance in breast cancer.

## 1. Introduction

Breast cancer is the most common malignancy in women. Surgery, radiation therapy, hormonal therapy, chemotherapy, and targeted therapy have greatly improved breast cancer patients’ survival. However, a high incidence of recurrence is common in aggressive and advanced diseases [1]. Drug resistance is a major cause of treatment failure in cancer chemotherapy. A better understanding of the mechanisms of chemotherapy failure is needed to improve current therapeutic methods and prediction of patients’ outcome.

Overexpression of ABC transporters has been proposed as a main cause of drug resistance, but targeting these drug efflux transporters did not show satisfactory clinical results [2]. Drug resistance is a multifactor phenomenon. Various mechanisms have shown to be involved. These mechanisms include changes in cellular responses, such as enhanced ability of DNA repair, stress toleration, or acquiring mechanisms to escape apoptosis [3,4]. In addition, there is growing evidence to support that antioxidant ability can contribute to chemotherapy resistance [5,6], such as a complex intracellular redox homeostasis system developed to adapt and protect cells against the harmful effects of oxidative stress [7]. NRF2 is a redox-sensitive factor [8] that controls transcriptional upregulation of antioxidant response element (ARE)-bearing genes, such as phase II detoxifying and oxidative stress enzyme genes [9]. NRF2 activity is in part regulated by KEAP1. In response to stress signals, KEAP1-bound NRF2 is released and translocated inside the nucleus of the cell for its transcriptional activity [9]. NRF2 can also be regulated independently by KEAP1. Phosphorylation of NRF2 by various protein kinases (PKC, GSK-3β, JNK, ERK, and Fyn) serve to promote nuclear accumulation, stability, activation, or degradation [10,11,12,13]. Epigenetic factors such as microRNAs [14,15] or NRF2 interaction with other proteins (nuclear recepotrs, caveolin-1) may also be involve in the regulation of NRF2 activity [16,17]. Upregulation of NRF2 has been reported in different types of cancer. A previous analysis of estrogen receptor (ER) positive breast cancer patients showed that high NRF2 expression is significantly associated with poor patient outcome [18]. Resistance to chemotherapeutic drugs is reported to be associated with alterations of the NRF2 signaling pathway and expression [19,20]. Downregulation of NRF2 activity by KEAP1 alteration increases 5-fluorouracil sensitivity in biliary tract cancer cells [19]. However, activation of NRF2 by small molecule activator tert-butylhydroquinone (tBHQ) also enhances resistance of neuroblastoma cells to cisplatin, etoposiside, and doxorubicin [20]. In Zhong et al. (2013) [21], higher expression of NRF2 has been found in doxorubicin resistant MCF-7 cells compared with doxorubicin sensitive MCF-7 parental cells. Antioxidant enzymes such as NQO1 are also expressed at higher levels in these drug resistant cells. These findings suggested that activation of NRF2 is associated with anthracycline resistance [21].

Several nuclear receptors have been reported to interact with NRF2 to regulate NRF2 transcriptional activity [16]. Repression of NRF2 pathway by nuclear receptors, such as estrogen receptor, retinoic acid receptors, and retinoid X receptor, has been shown in breast cancer cells [16,22,23]. For example, ERα binds to NRF2 and inhibits NRF2 transactivation [24,25], whereas RXRα blocks transactivation by binding directly to the Neh7 domain of NRF2 [26]. Notably, the nuclear receptor co-repressor 2 (NCOR2) has been demonstrated to inhibit NRF2 transcription upon activation of glucocorticoid receptor [27]. Our group has previously identified a novel splice variant of NCOR2, BQ323636.1 (called BQ in short) from SpliceArray profiling of breast cancer [28]. BQ is derived from alternative splicing of NCOR2. We have found that BQ is significantly associated with tamoxifen resistance through compromising the suppressive role of NCOR2 in regulating estrogen-receptor element activity, rescuing the transcriptional suppression of tamoxifen on ERα target genes [28,29].

We have previously uncovered that BQ can serve as a robust predictor for tamoxifen resistance in ER positive breast cancer. NCOR2 is a key co-repressor protein that functions to modulate transcription of various transcription factors [30]. We have previously shown that BQ as a truncated spice variant of NCOR2, can compete with NCOR2 resulting in a defective functional co-repressor complex [29]. Given the role of NCOR2 inhibiting NRF2 activity, and that BQ can compete with NCOR2 to modulate co-repressor function on NRF2 activity, we hypothesized BQ can modulate oxidative stress by enhancing NRF2 activity, and therefore contribute to the development of chemoresistance.

## 2. Results

### 2.1. Epirubicin Resistant Cells Are More Refractory to Oxidative Stress

Chemotherapeutic drugs, such as anthracyclines and platinum compounds, induce cellular damage and cell death in cancer cells partly via the production of reactive oxygen species (ROS) [31]. However, the cytotoxic effect of a chemotherapeutic drug is often compromised in cells with hyperactive antioxidant signaling. Therefore, factors that enhance antioxidant signaling might be able to compromise the effect of chemotherapeutic drugs and promote chemoresistance.

To test this hypothesis, we first compared the ROS levels in the parental MCF-7 epirubicin-resistant (MCF-7 EpiR) cells [32] with the parental MCF-7 breast cancer cells. To this end, the ROS levels were determined by CellROX Deep Red Reagent, which is a fluorogenic probe for measuring cellular oxidative stress (Appendix A). The results showed that MCF-7 EpiR cells had lower steady state ROS levels (Figure 1A). On treatment with oxidant tert-Butyl hydroperoxide (tBHP), although the parental MCF-7 cells were sensitive to oxidative stress, MCF-7 EpiR cells were comparably more refractive (Figure 1A). The results suggested that resistant cells have higher anti-oxidative stress capacities. Through SRB and clonogenic assays, we confirmed that the MCF-7 EpiR cells had enhanced cell viability under cytotoxic oxidative conditions (Figure 1B,C) (Appendix A).

Likewise, on treatment with epirubicin, MCF-7 EpiR cells were more refractive to epirubicin-induced ROS than MCF-7 cells (Figure 2A). The transcription factor NRF2 is a master regulator of oxidative stress and reported to be associated with chemoresistance [33,34]. As antioxidant response element (ARE) is the main transcription binding site recognized by NRF2, an ARE luciferase reporter assay was used to study NRF2 transcription activity (Appendix A). We found that MCF-7 EpiR cells displayed higher ARE-associated luciferase activity than MCF-7 cells. This indicated that MCF-7 EpiR cells have higher NRF2 transcriptional activity. Knockdown of NRF2 by specific siRNAs (Appendix A) in MCF-7 EpiR cells resensitized these cells to epirubicin (Figure 2B) (Appendix A), reconfirming the role of NRF2 in chemoresistance. Interestingly, while we observed that NRF2 expression is enhanced in epirubicin resistant cells (Figure 2C), results from subcellular fractionation failed to demonstrate enhanced nuclear NRF2 expression in the resistant cells (Appendix A), suggesting other mechanisms are involved in the regulation of NRF2 transcription activity in these resistant cells.

### 2.2. BQ Is Associated with Chemoresistance of Breast Cancer

Given that NCOR2 can inhibit NRF2 activity [27], and that BQ can compete with NCOR2 to regulate ERα transcriptional activity [28,29], we hypothesized that BQ may modulate oxidative stress by enhancing NRF2 activity, and therefore contribute to the development of chemoresistance.

To determine whether BQ may contribute to epirubicin resistance in breast cancer cells, we first compared the expression of BQ in drug-sensitive cells and resistant cells. Western blot was used to determine the expression levels of BQ and NCOR2 in the MCF-7 EpiR, MCF-7 TaxR and parental MCF-7 breast cancer cells. As expected, although BQ levels were higher in the resistant cells, NCOR2 expression was comparatively lower (Figure 3A), suggesting that high BQ and low NCOR2 expression levels is associated with chemoresistance. To test this conjecture, we next performed immunohistochemical (IHC) staining of BQ on a tissue microarray (TMA) constructed from a cohort of 62 patients with records of chemotherapeutic treatment. Representative images of BQ IHC staining are shown in Figure 3B. Of these the cases with analyzable data, 16 had subsequently suffered cancer relapse, whereas 43 had not. Although the number of cases is relatively small, both Mann–Whitney and Chi square test nevertheless showed statistically significant higher nuclear BQ expression in patients who suffered a relapse (* *p* = 0.018 and ** *p* = 0.001, respectively) (Figure 3C). In addition, and as expected, nuclear BQ expression was enhanced in epirubicin resistant cells (Appendix A). These in vivo and in vitro data support our hypothesis that BQ contributes to chemoresistance in breast cancer and its high nuclear expression might be associated recurrence after chemotherapy.

### 2.3. BQ Overexpression Contributes to Epirubicin Resistance

We next employed clonogenic assay to determine whether BQ overexpression would alter the sensitivity of breast cancer cells to epirubicin. We established two stable BQ-overexpressing cell models from MCF-7 and ZR-75 cell lines (MCF-7 BQ and ZR-75 BQ), respectively. In these cells, BQ overexpression was confirmed by both Qpcr and Western blot (Appendix A). The parental and BQ-overexpressing cells were treated with various doses of epirubicin for 14 days. We found that BQ overexpression conferred epirubicin resistance to both MCF-7 and ZR-75 cells (Figure 4A). Similarly, BQ overexpression could protect cells against cytotoxic oxidative conditions from the results of SRB assays and clonogenic assay with increased cell viability seen in BQ overexpressing cells treated with tBHP (Appendix A). To further illustrate that BQ overexpression could protect cells from oxidative stress, ROS levels were measured in BQ overexpressing and parental cells upon epirubicin treatment (Figure 4B). The results showed that epirubicin treatment induced ROS levels significantly in the empty expression vector harboring cells. The addition of antioxidant NAC reduced the levels ROS induced by epirubicin, suggesting that ROS induced by epirubicin can be depleted by antioxidant. In comparison, in BQ overexpressing cells, the level of ROS induced by epirubicin was significantly reduced. These results suggest that ectopic expression of BQ could limit ROS accumulation induced by chemotherapy.

Consistently, treatment of breast cancer cells with epirubicin induced cellular apoptosis, as revealed by Annexin V Alexa Fluor 647 and SYTOX Blue Dead cell stain double-staining (Appendix A). Results showed that BQ overexpression could significantly reduce epirubicin-induced apoptosis. (Figure 4C). Similarly, combined treatment of NAC and epirubicin in BQ overexpressing cells did not show significant reduction of apoptotic cells compared to control cells, indicating that epirubicin-induced ROS production is involved in epirubicin-induced apoptosis in these cells (Figure 4C). Although epirubicin treatment induced cleavage of apoptotic markers PARP and Caspase 3, we also found reduced cleavage of these apoptotic markers in BQ-overexpressing cells (Appendix A). These results suggested that overexpression of BQ could counteract cellular apoptosis induced by chemotherapy via induced ROS generation, thus reducing the cytotoxic effects of the drugs.

Our in vitro results suggested that overexpression of BQ contributes to epirubicin resistance. The involvement of BQ in epirubicin resistance was further confirmed using an in vivo mouse xenograft model (Figure 4D). We examined the dosage effect of epirubicin on the tumors derived from control and BQ-overexpressing cells. ZR75-empty vector (EV) tumors displayed significant reductions in tumor growth when treated with 2 mg/kg and 5 mg/kg of epirubicin (Figure 4D(i,ii)), indicating that ZR75 EV tumors were sensitive to epirubicin compared with untreated control. In contrast, tumors with BQ overexpression did not respond to the same range of epirubicin treatment (Figure 4D(iii,iv)). Taken together, we confirmed that BQ overexpression can confer resistance to epirubicin treatment in breast cancer.

### 2.4. BQ Overexpression Upregulates the Expression of Antioxidant Enzymes

Given that epirubicin resistant cells are refractory to oxidative stress and that BQ overexpression might contribute to epirubicin resistance, we hypothesized that BQ overexpression could promote the anti-oxidative stress activity of breast cancer cells and thus epirubicin resistance. NRF2 is a master regulator of oxidative stress, controlling the expression of anti-oxidative genes [33]. We have shown previously that BQ can bind to NCOR2 to form a defective co-repressor complex to compromise the suppressive role of NCOR2 [29]. As NCOR2 is a transcriptional repressor of NRF2, it is very likely that BQ overexpression protect cells from oxidative stress through promoting the activity of NRF2.

To probe the link between BQ and NRF2, we evaluated the expression levels of the well-characterized NRF2 downstream antioxidant targets NQO1, HMOX1, GCLC, and GSTP1 by RT-qPCR. The results showed that BQ overexpression led to the upregulation of NQO1, HMOX1 and GCLC at the transcriptional level (Figure 5A) and NQO1 and HO-1 (gene name: HMOX1) at protein level in both MCF-7 and ZR75 (Figure 5B). Similarly, epirubicin resistant cells also displayed enhanced expression of NQO1 and HO-1 (Appendix A). To further validate our findings in vivo, we performed immunohistochemical (IHC) staining of BQ and NQO1 on TMA constructed from 124 cases of primary breast cancers. Representative images of NQO1 IHC staining were shown in Figure 5C. Nuclear expression of BQ and NQO1 in these clinical samples was scored semiquantitatively as previously described [29]. We found that NQO1 protein expression was positively and significantly correlated with BQ expression (R^2^ = 0.4555; *p* = 0.0173) (Figure 5D). These results provide in vivo evidence to support our in vitro findings that BQ overexpression upregulates NQO1 expression. Moreover, consistent with our hypothesis, knockdown of NRF2 or NQO1 (Appendix A) in BQ overexpressing cells resensitized the cells to epirubicin treatment (Figure 5E). This further suggests that the effect of BQ on chemoresistance is mediated via NRF2 and NQO1.

### 2.5. BQ Overexpression Does Not Affect NRF2 Expression or Post-Translational Modification

As the transcriptional activity of NRF2 may be regulated by different mechanisms, including its expression level [35], subcellular localization [36], and post-translational modifications, such as Ser-40 phosphorylation, which facilitates the release of NRF2 from its repressor [37] and co-repressors [27], we examined the effect of BQ overexpression on the expression and activity of NRF2. Results showed that BQ overexpression did not alter expression of NRF2, at both mRNA (Figure 6A) and protein levels (Figure 6B), nor alter the phosphorylation status of NRF2 Ser-40 in both cell lines (Figure 6B). Nucleocytoplasmic fractionation showed that BQ overexpression could not enrich NRF2 in the nuclear fraction (Appendix A). We also examined the expression of co-repressor NCOR2, a negative regulator that can inhibit the activity of NRF2 [27]. Both qPCR (Figure 6A) and Western blot results (Figure 6B) confirmed that BQ overexpression could not alter the expression of NCOR2.

### 2.6. BQ Modulates the Ability of NCOR2 to Suppress Transcription on Antioxidant Response Element

NCOR2 forms a homodimer in an antiparallel fashion (i.e., N-terminal binds to C-terminals) to serve as a dock for further recruitment of other co-repressor proteins in the co-repressor complex [38]. As a truncated NCOR2 splice variant, BQ retains the only N-terminus of wild type NCOR2 for dimerization with NCOR2. In agreement with this, we have previously demonstrated that BQ can interact with NCOR2 resulting in defective co-repressor activity on the estrogen receptor [29]. As it has been reported that NCOR2 is a co-repressor for transcriptional activity of NRF2 [27], we thus speculated that overexpression of BQ can also result in defective co-repressor activity of NCOR2 on NRF2 ARE element.

To test this hypothesis, we transiently overexpressed NCOR2 in BQ overexpressing cells (Appendix A) and measured the luciferase activity driven by antioxidant response element (ARE), an NRF2 response element on the promoter (Figure 7A). We found that overexpression of NCOR2 alone could suppress ARE driven-luciferase activity, whereas co-expression of NCOR2 with BQ partially released such repression. Therefore, BQ overexpression can compromise the suppressive effect of NCOR2 on the transcriptional activity of NRF2. Through co-immunoprecipitation (Figure 7B), we confirmed that NCOR2 could indeed interact with NRF2 in both MCF-7 and ZR-75 cells, but that such an interaction was significantly reduced in BQ overexpressing cells. Furthermore, we found BQ was enriched in the NCOR2-immunoprecipitates. This suggested that BQ could interact with NCOR2 to sequester NCOR2 from its interaction with NRF2. Thus, our results confirm that BQ overexpression enhances NRF2-mediated gene expression via inhibiting the interaction between NCOR2 and NRF2.

## 3. Discussion

Anthracyclines is one of the most active and widely used chemotherapeutic agents for breast cancer [39]. The development of chemoresistance poses difficulties to cancer therapy as chemoresistant cancer cells become more difficult to eliminate using the same chemotherapeutics. Even though patients might have high initial response rates, they might eventually relapse due to acquired resistance. Therefore, chemoresistance reduces the number of treatment options for advanced diseases. Most anticancer drugs can induce the formation of ROS, which triggers cell apoptosis via the mitochondrial pathway initiated by cytochrome c release or activation of death receptor cascade [31,40]. Altered redox signaling and oxidative stress response could contribute to chemoresistance. Several cellular defense mechanisms against oxidative stress have been shown to be upregulated in cancer cells by NRF2. They include the thioredoxin/thioredoxin reductase (Trx/TrxR) system, peroxiredoxins (Prxs), and several glutathione S-transferases (GSTs) [41]. Consistently, constitutive NRF2 activation can protect cancer cells from chemotherapy [21]. Aberrant NRF2 activation has been reported [42] to be mediated by a number of mechanisms, including somatic mutation of NRF2 and KEAP1 [43], DNA methylation of KEAP1 gene promoter [44], p62 accumulation which compete with NRF2 for KEAP1 [45] and transcriptional activation of NRF2 gene by other oncogenic factors (such as K-Ras, B-Raf, Myc, and mutant p53) [46,47,48,49]. However, the underlying molecular mechanism by which NRF2 transcriptional activity is enhanced to confer drug resistance in breast cancer cells is not well understood.

Our study provides evidence to support the idea that BQ can induce chemoresistance in breast cancer by altering the transcriptional suppressive function of NCOR2 on NRF2. Based on our current findings and previous findings [29], we propose a mechanism whereby BQ can contribute to chemoresistance via modulation of oxidative stress (Figure 8). In this model, in the absence of BQ, NCOR2 self-dimerizes to form a central platform to recruit other corepressor components to form a functional NCOR2 corepressor complex. This repressive complex binds to NRF2 and suppresses its transactivation of target antioxidant genes via the ARE (Figure 8A). Conversely, in the presence of BQ, BQ complexes with NCOR2 and sequesters it from forming transcriptional suppressive complex with NRF2. This leads to defective co-repressor activity of NCOR2 on NRF2 ARE element, and increased induction of ARE-mediated transcription of antioxidant enzymes (Figure 8B). The enhanced expression of antioxidant enzymes can eliminate the excessive cytotoxic ROS induced by chemotherapeutic drugs. Thus, BQ overexpression can provide survival advantages to breast cancer cells under chemotherapy. Based on the role of BQ in modulating oxidative stress, a novel oncogenic role for BQ in contributing to epirubicin resistance is proposed. This current study strengthens the potential value of BQ not only as a biomarker for tamoxifen resistance in breast cancer but also could serve as a potential predictor of chemoresistance.

Alternative splicing has recently been viewed as a new hall mark of cancer [50]. Research has now uncovered various alternative spliced isoforms implicated in breast cancer, including BRCA1, DMTF1, FGFR, HER2, KLF6, Survivin, and TP53 [51]. Abnormal expression of splice factors in cancer changes the alternative splicing of pre-mRNAs. Deregulation of these splicing factors has been linked to breast cancer development [52]. More than 30 human splicing factors have now been defined, and some of them are characterized as either proto-oncogenes, such as SRSF1, SRFS3, DAM1, and PELP1, or tumor suppressors, such as RBM and PRMT6, in breast cancer [52]. From our result in Figure 3A, the ratio of BQ to NCOR2 was altered in epirubicin resistant breast cancer cells. As BQ is an alternative splicing product of NCOR2, our findings also underscore the fact that deregulation of splicing factors might induce drug resistance. However, the factors for regulation of BQ production are as unclear. Once these specific splicing factor(s) responsible for BQ production is identified, targeting such factors could be a new therapeutic approach for breast cancer.

In summary, this study has characterized a new molecular mechanism involved in the regulation of NRF2 mediated by BQ, and provides a better understanding of the molecular basis for the regulation of oxidative stress and the development of chemoresistance in breast cancer. Our present work also suggests that BQ could be a novel therapeutic target as well as a predictive marker for chemoresistance.

## 4. Materials and Methods

### 4.1. Cell Culture

MCF-7 and ZR-75-1 were purchased from the American Type Culture Collection (ATCC) and were re-authenticated previously [29]. MCF-7 cells were cultured in Dulbecco’s Modified Eagle’s Medium (DMEM) (Gibco, Grand Island, NY, USA) supplemented with 1% penicillin–streptomycin (Gibco, USA) and 10 % fetal bovine serum (FBS) (Gibco, USA). ZR-75-1 cells were cultured in Improved Minimum Essential Medium (IMEM) (Gibco, USA) supplemented with 1% penicillin–streptomycin and 10% FBS. Epirubicin resistant MCF-7 cells (MCF-7 EpiR) and paclitaxel resistant MCF-7 cells (MCF-7 TaxR) were previously established by our group and were maintained in 10 µM epirubicin and 50 nM paclitaxel, respectively [32]. Stable BQ-overexpressed ZR-75 and its control ZR-75 EV cell lines were previously generated by our group using lentiviral system [29]. The cells were cultured in IMEM supplemented with 1% penicillin–streptomycin, 10 % FBS, and 5 µg/mL puromycin (Gibco, USA). The stable BQ-overexpressed MCF-7 cells were generated by transfecting pCDNA3.1-BQ-His (MCF-7 BQ) or pCDNA3.1-Empty (MCF-7 EV), followed by geneticin (Gibco, USA) selection. The stable cell lines were cultured in DMEM supplemented with 1% penicillin–streptomycin, 10 % FBS and 500 µg/mL geneticin.

### 4.2. Plasmids Transfection

The plasmid pcDNA3.1-BQ-His has been previously described [28]. NCOR2 expression plasmid pCMV6-NCOR2 and its empty vector pCMV6-Entry were purchased from Origene (Rockville, MD, USA). Luciferase reporter vectors pGL3-basic and pRL-SV40 were purchased from Promega (Madision, WI, USA). Antioxidant response element (ARE) of human *NQO1* gene has been identified previously [53]. Primers designed were aligned with the *NQO1* promoter containing human *NQO1* ARE sequence and added cutting sites for Nhe1 and Xho1 restriction enzymes, forward primer: 5′-CTA GCT AGC TAG CCG AGT AGC TGG GAC TTA CAG G-3′; reverse primer: 5′-CCG CTC GAG CGG GCA CGA AAT GGA GCA GAA AAA GAG C-3′. Amplified sequence was cloned into pGL3-basic, i.e., pGL3-ARE. Expression plasmid transfections were performed with Lipofectamine 2000 (Invitrogen, Carlsbad, CA, USA) or Lipofectamine 3000 (Invitrogen, Carlsbad, CA, USA) according to the manufacturer’s recommendations.

### 4.3. Small Interfering RNA (siRNA) Transfection

For gene knockdown, cells were transiently transfected with ON-TARGETplus siRNA Reagents purchased from Dharmacon (Lafayette, CO, USA) using DharmaFECT 1 transfection reagent (Dharmacon, Lafayette, CO, USA) according to the manufacturer’s instructions. siRNAs reagents used were: NRF2 siRNA (L-003755-00), NQO1 siRNA (L-005133-00), and the non-specific control siRNA ON-TARGETplus Non-targeting Pool (D-001810-10) was confirmed to have minimal targeting of human genes.

### 4.4. Luciferase Reporter Assay

Cells were co-transfected with pGL3-ARE and Renilla luciferase control plasmid pRL-SV40 using Lipofectamine 3000 (Invitrogen, Carlsbad, CA, USA). For promoter analysis, cells were collected 24 h post-transfection, washed twice in PBS, and harvested for Firefly/Renilla luciferase assays using the Dual-Luciferase reporter assay system (Promega, Madision, WI, USA) according to manufacturer’s instruction. Luminescence was then read using Infinite 200 Pro microplate plate reader (TECAN, Mannedorf, Switzerland).

### 4.5. Real-Time Quantitative PCR

Total RNA was extracted with the RNeasy Mini Kit (Qiagen, Hilden, Germany) following the manufacturer’s protocol. Up to 1000 ng total RNA was reverse transcribed into cDNA by PrimeScript RT Reagent Kit (Takara, Kusatsu, Japan) following the manufacturer’s protocol. SYBR green-based real-time PCR reaction was carried out using ABI 7900HT Real-Time PCR machine (Applied Biosystems, Foster City, CA, USA). Relative expression levels were determined using the 2^−ΔΔCt^ method. Primer sequences used in this study were listed in Appendix A.

### 4.6. Western Blot Analysis

Cell lysates were prepared for Western blot analysis as previously described [54]. The proteins were probed with primary antibody against BQ323636.1 [29], NCOR2 (#ab24551, Abcam, Cambridge, UK), NRF2 (#NBP1-32822, Novus biologicals, Centennial, CO, USA), and phosphor-NRF2 (S40) (#ab76026, Abcam, Cambridge, UK). β-Actin (#sc-47778), Lamin B (#sc-6216) and NQO1 (#sc-32793) antibodies were purchased from Santa Cruz Biotechnology (Santa Cruz, CA, USA). Antibodies against β-Tubulin (#86298S), DYKDDDDK Tag (#2368S) and HO-1 (#5853S) were purchased from Cell Signaling Technology (Danvers, MA, USA). PARP (#13371-1-AP) and Caspase 3 (#66470-2-Ig) antibodies were obtained from Proteintech group (Rosemont, IL, USA). Anti-His tag antibody was obtained from Wako (#010-21861, Osaka, Japan). Primary antibodies were detected using anti-mouse (#P0447, Dako, Glostrup, Denmark), anti-rabbit (#P0448, Dako, Glostrup, Denmark), or anti-goat HRP (#ab6741, Abcam, Cambridge, UK) conjugates as appropriate and visualised using the ECL detection system (Bio-Rad, Hercules, CA, USA). The original blots (Appendix A) were scanned and densitometry readings/intensity ratio of each band calculated for analysis (Appendix A).

### 4.7. Sulforhodamine B (SRB) Assay

A total of 4000 cells were seeded in each well of a 96-well plate. Cells were treated with tert-butyl hydroperoxide (tBHP) (Sigma-Aldrich, St Louis, MO, USA) the next day. After culture for 1 to 5 days, cells were fixed with 100 μL of trichloroacetic acid Sigma-Aldrich, St Louis, MO, USA) for 1h at 4 °C. Plates were then washed five times with running water and 100 μL of SRB solution (0.4% SRB in 0.1% acetic acid) was added. The plates were stained for 1 h at room temperature and rinsed three times with 1% acetic acid and air-dried. One-hundred microliters of 10 mM Tris base solution was then added to each well to dissolve the protein-bound dye, and optical density at 492 nm was measured in a microplate reader (TECAN, Mannedorf, Switzerland).

### 4.8. Clonogenic Assay

A total of 1000 cells were seeded into each well of the six-well plates. Cells were then treated with tBHP or epirubicin (EBEWE, Unterach, Austria) and grown for 14 days until visible colonies formed. Colonies were rinsed three times with PBS and fixed with 4% PFA for 20 min at room temperature. The colonies were stained with 0.5% crystal violet (Sigma-Aldrich, St Louis, MO, USA) and air-dried. Two milliliters of 33% acetic acid was used to dissolve bound crystal violet. Optical density was then measured at 492 nm using a microplate reader (TECAN, Mannedorf, Switzerland).

### 4.9. ROS Measurement by Flow Cytometry

Detection of ROS in cells was done by CellROX Deep Red Flow Cytometry Assay Kit (Molecular Probes, Eugene, OR, USA) according to manufacturer’s instructions. The CellROX Deep Red reagent is a cell permeable dye that does not fluoresce in a reduced state while it becomes fluorescent upon oxidation by ROS. The emission maxima of CellROX Deep Red is approximately 665 nm and is measurable by flow cytometry. Briefly, cells were treated with epirubicin (EBEWE, Unterach, Austria). After treatment, cells were trypsinized and counted by hemocytometer. 1 × 10^6^ cells/mL were resuspended in cell culture medium (without phenol red). Cells were stained with CellROX Deep Red reagent in the 37 °C incubator for 1 h. The samples were immediately analyzed by flow cytometry by LSR Fortessa analyser (BD Biosciences, NJ, USA). The data were analyzed by the FlowJo analysis software (FlowJo LLC, Ashland, OR, USA) and the values for mean fluorescence intensity, which represent the relative ROS levels, were obtained.

### 4.10. Flow cytometric Determination of Apoptosis

Cells were analyzed by Annexin V Alexa Fluor 647 stain (Invitrogen, Carlsbad, CA, USA) plus SYTOX Blue Dead cell stain (Invitrogen, Carlsbad, CA, USA) double-staining method according to manufacturer’s protocol. The cells after designed treatments were collected and subjected to the analysis. A minimum of 10,000 events were then analyzed by LSR Fortessa analyzer (BD Biosciences, NJ, USA) with FlowJo analysis software (FlowJo LLC, Ashland, OR, USA) for acquisition and analysis.

### 4.11. Immunoprecipitation

Cell were lysed in IP buffer (1% NP-40, 150 mM NaCl, 50 mM Tris-HCl pH 7.4, 10 mM NaF, 1 mM Na_3_VO_4_, 10 mM N-ethyl-amide and protease inhibitors [Complete protease inhibitor cocktail; Roche, Mannheim, Germany]) and precleared with 30 μL of Dynabeads Protein A (Invitrogen, Carlsbad, CA, USA) for 4 h at 4 °C. Protein concentration of the precleared lysates were determined and the lysate was immunoprecipitated with the indicated antibodies or IgG negative control (Dako, Glostrup, Denmark) for overnight at 4 °C. Forty microliters of Dynabeads Protein A was added to the lysates and incubated for a further 4 h at 4 °C. The beads were washed five times with ice-cold PBS, followed by boiling at 100 °C in 2× SDS loading dye for 10 min to elute the proteins. Eluted proteins were analyzed by Western blotting. The intensities of the protein bands were quantified by ImageJ software. Ratio of (IP: NCOR2) to (INPUT) was calculated. This value represented the relative bindings of NCOR2 to NRF2 at protein level.

### 4.12. Subcellular Fractionation

Subcellular fractionation was performed using the NE-PER^TM^ Nuclear Cytpoplasmic Extraction reagents (Thermo Scientific, Rockford, IL, USA) following manufacturers’ protocol. Protein concentrations of the cytoplasmic and nuclear extracts were determined. The extracts were then mix and boiled with 5X SDS loading dye for subsequent Western blotting analysis.

### 4.13. Orthotopic Mouse Model

Female nude mice of 5–6 weeks old were used for this study. On the day of inoculation, approximately 1 × 10^7^ ZR-75 EV or ZR-75 BQ cells were injected to the abdominal mammary fat pad of the mouse. When the tumors were palpable, these mice were randomized into epirubicin treatment and untreated control groups. The treatment groups received epirubicin (EBEWE, Unterach, Austria) diluted in PBS by intraperitoneal injection, whereas the control group received PBS only. Tumor sizes were measured regularly using calipers and the tumor volumes were calculated as longest diameter × (shortest diameter)^2^/2. At the end of experiment, mice were euthanized, and tumors harvested. All the procedures were reviewed and approved by HKU Committee on the Use of Live Animals in Teaching and Research (CULATR No: 5103-19).

### 4.14. Tissue Microarray

One-hundred-and-forty-one cases of breast cancer diagnosed between the years 1993 to 2003 with clinical follow up data were retrieved from the records of the Department of Pathology, Queen Mary Hospital of Hong Kong, with approval (UW 08-147) by the Institutional Review Board of The University of Hong Kong. Histological sections of these cases were reviewed by the pathologist, the representative paraffin tumor blocks were chosen as donor block for each case. Selected areas were marked for construction of tissue microarray (TMA) blocks. A total of 124 could be assessed and scored for both BQ and NQO1 expression (Appendix A). Similarly, 62 cases of breast cancer diagnosed between the years 1993 to 2003 with clinical follow up data and with record of having received chemotherapeutic treatment were retrieved and analyzed for BQ expression and relapse status (Appendix A).

### 4.15. Immunohistochemistry

The immunohistochemistry procedures were described previously [29]. TMA sections were deparaffinized and rehydrated with repeated incubation of ethanol and xylene; 0.01 M citrate buffer (pH 6) was used for antigen retrieval and the slides were then immersed in 3% H_2_O_2_ (in methanol) at room temperature for quenching endogenous peroxidase. The slides were then rinsed in 1× PBS with 0.05% Tween-20 (PBST) twice, followed by incubation with primary antibodies at 4 °C overnight. Antibodies used were anti-BQ323636.1 (1:50) [29] and anti-NQO1 (1:200) (#sc-32793, Santa Cruz, CA, USA). The slides were washed with 1 x PBST and incubated with EnVision+ System- HRP labelled Polymer Anti-Rabbit (Dako, Glostrup, Denmark) at room temperature for 30 min in dark. The excess reagent was washed off with 1× PBST and Chromogen DAB/substrate reagent (Dako Glostrup, Denmark) was added to the slides and incubated for 6 min. The slides were dehydrated and mounted. To visualize and assess protein expressions, Aperio ScanScope system (Aperio technology, USA) was used. TMA slides were scanned, and individual TMA spots were assessed using Aperio’s image viewer, ImageScope. The intensities and percentages of BQ and NQO1 nuclear staining were scored as previously described [29] by two independent individuals.

### 4.16. Statistical Analysis

Statistical analysis was conducted GraphPad Prism version 6 (GraphPad Software Inc, San Diego, CA, USA) and SPSS version24 (IBM, Armonk, NY, USA). Results were expressed as mean ± SD from at least three independent experiments. Statistical differences between two groups’ means were evaluated by two-tailed Student’s *t*-test. Two-way analysis of variance (ANOVA) was used between groups of 2 variables. Significant difference was considered when *p*-value: * *p* < 0.05, ** *p* < 0.01 and *** *p* < 0.001. The correlation between BQ323636.1 with relapses was analyzed by Mann-Whiney U Rank test analysis. *p* < 0.05 was considered statistically significant. The data was also dichotomized into two groups for protein expression, as high or low expressers with median expression level as the cutoff. The correlations were analyzed by chi square test. The correlation between BQ323636.1 and NQO1 expression was analyzed by linear regression analysis. *p* < 0.05 was considered statistically significant.

## Figures and Tables

**Figure 1 cancers-12-00533-f001:**
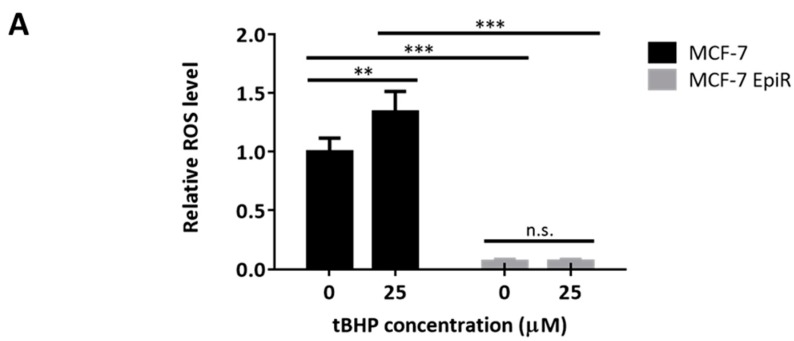
Epirubicin-resistant breast cells have enhanced capacity against oxidative stress. (**A**) Epirubicin-sensitive MCF-7 and Epirubicin-resistant MCF-7 EpiR cells were treated with 25 µM of oxidative stress inducer tert-Butyl hydroperoxide (tBHP) for 24 h. Epirubicin-sensitive MCF-7 and Epirubicin-resistant MCF-7 EpiR cells were stained with CellROX Deep Red reagent and analyzed for ROS levels using flow cytometry. Data were analyzed by FlowJo software. The mean fluorescence values were presented as relative ROS level compared to untreated cells (0 µM). Data presented as means ± SD and Student’s *t*-test was used to compare the means: * *p*  <  0.05; ** *p*  < 0.01; *** *p*  <  0.001; n.s. nonsignificant. (N = 3) (**B**) SRB assay was used to analyze the cell viability of MCF-7 EpiR cells treated with tBHP as compared to MCF-7 cells. Relative cell viability under different concentrations of tBHP treated for 72 h as compared with untreated cells. Data was presented as means ± SD and analyzed by two-way ANOVA and found to be significantly different between MCF-7 and MCF-7 EpiR cells, * *p* < 0.05. (N = 3) (**C**) MCF-7 and MCF-7 EpiR cells were treated with increasing concentrations of tBHP, and their sensitivity to tBHP was assessed by clonogenic assay. Their clonogenicity in response to tBHP was analyzed by two-way ANOVA and found to be significantly different (** *p* < 0.01) from one another (N = 3).

**Figure 2 cancers-12-00533-f002:**
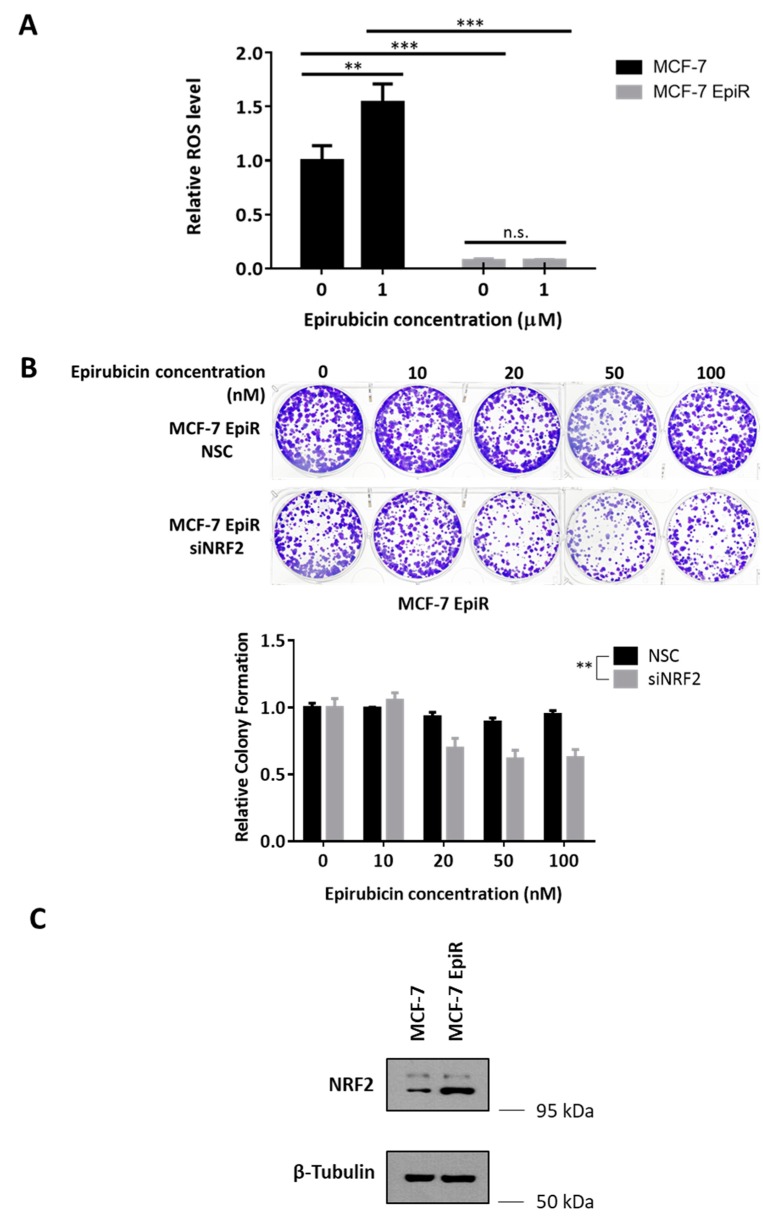
NRF2 modulates epirubicin resistance in breast cancer cells. (**A**) MCF-7 and MCF-7 EpiR cells were treated for 24 h with epirubicin at 1 µM. MCF-7 and MCF-7 EpiR cells were stained with CellROX Deep Red reagent and analyzed for ROS levels by flow cytometry. Data were analyzed by FlowJo software. The mean fluorescence values were presented as relative ROS level compared to untreated cells (0 µM). Data presented as mean ± SD. Student’s *t*-test was used to compare the means: ** *p* < 0.01; *** *p* < 0.001; n.s. nonsignificant. (N = 4) (**B**) Knockdown of NRF2 was achieved by transfecting 4 specific siRNA against NRF2 (siNRF2, 150pmol) to MCF-7 EpiR cells in a 6-well plate. Non-targeting siRNAs were used as control (NSC). At 24 h post-transfection, these cells were seeded in 6-well plates and treated with increasing doses of epirubicin for 14 days. Their sensitivity to epirubicin was assessed by clonogenic assay. Their clonogenicity in response to epirubicin was analyzed by two-way ANOVA and found to be significantly different (** *p* < 0.01) from one another. (N = 3) (**C**) Expression of NRF2 in MCF-7 cells and MCF-7 EpiR cells was detected by Western blot. β-Tubulin served as the loading control. (N = 3).

**Figure 3 cancers-12-00533-f003:**
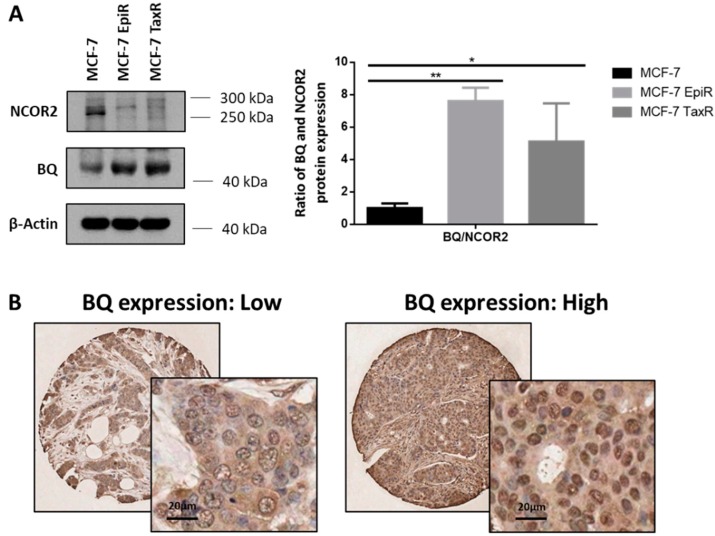
BQ is associated with chemoresistance of breast cancer. (**A**) Expression of NCOR2 and BQ in MCF-7, MCF-7 EpiR, and MCF-7 TaxR cells were detected by Western blot. β-Actin served as the loading control (left panel). The intensities of NCOR2 and BQ bands were quantified by Image J analysis software which represents their relative protein expressions. Ratios of relative BQ to NCOR2 expression were calculated (right panel). Data presented as means ± SD, * *p* < 0.05; ** *p* < 0.01. (N = 3) (**B**) Immunohistochemical (IHC) staining of BQ in primary breast cancer samples in tissue microarray. Representative images of low or high BQ nuclear expression were shown. (**C**) Nuclear BQ expression score was associated with relapse status (Yes for with relapse; No for without relapse) as demonstrated with Mann–Whitney test in primary breast cancer samples; * *p* = 0.018. Nuclear BQ expression score were dichotomized at the median value and Chi square test performed for association with relapse status (Chi-square test *p* = 0.001).

**Figure 4 cancers-12-00533-f004:**
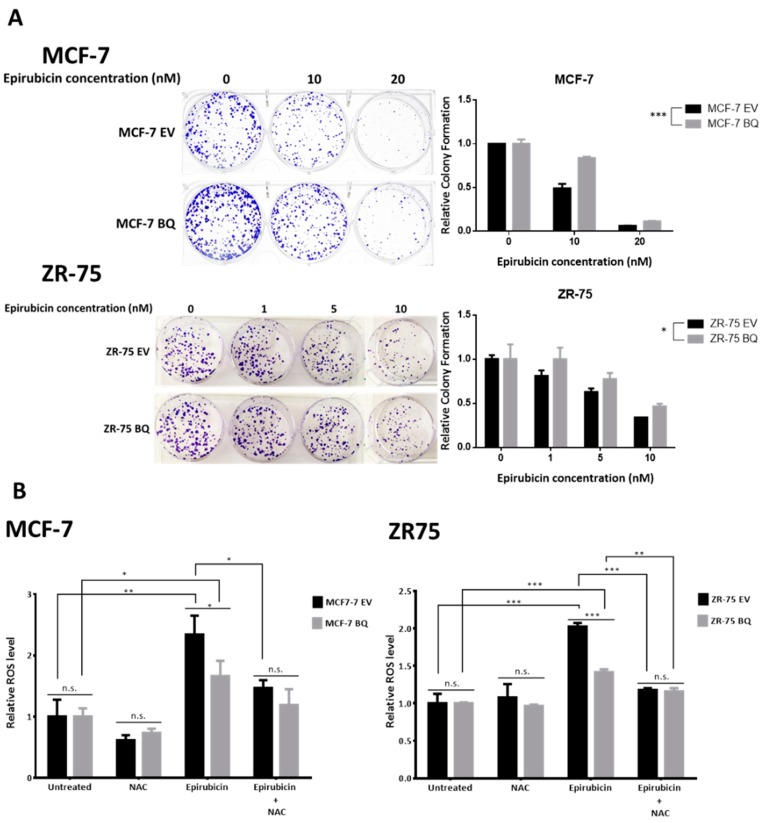
BQ overexpression contributes to epirubicin resistance. (**A**) Clonogenic assay was performed to assess the effect of epirubicin on the cell viability of control cells and BQ overexpressing cells. Cells were treated with different doses of epirubicin for 14 days. The cells were stained with crystal violet (left panel). The results (right panel) represent the average of three independent experiments. Data presented as mean ± SD and analyzed by two-way ANOVA (* *p* < 0.05; *** *p* < 0.001). (N = 3) (**B**) The effect of antioxidant N-Acetyl Cysteine (NAC) on the level of ROS induced by epirubicin. BQ-overexpressing cells or control cells were treated for 24 h with 5 µM NAC, 1 µM epirubicin alone or combined. Untreated cells served as control. Cells were stained with CellROX Deep Red reagent and analyzed for ROS levels by flow cytometry. Data were analyzed by FlowJo software. The mean fluorescence values were represented as relative ROS level compared to untreated control cells. (Student’s *t*-test, mean ± SD, * *p* < 0.05; ** *p* < 0.01; *** *p* < 0.001; n.s. nonsignificant) (N = 3) (**C**) The effect of NAC on apoptosis induced by epirubicin. BQ overexpressing cells or control cells were treated for 24 h with 5 µM NAC, 1 µM epirubcin alone or combined. Untreated cells served as control. Apoptosis was measured by Annexin V Alexa Fluor 647 stain plus SYTOX Blue Dead cell stain staining, followed by flow cytometry analysis. Representative cytofluorimetric plots were shown in Appendix A. Bars show the percentages of cells that were apoptotic cells. Results are expressed as mean ± SD. (Student’s *t*-test, mean ± SD, ** *p* < 0.01; *** *p* < 0.001; n.s. nonsignificant) (N = 3) (**D**) Tumor xenograft models in mouse used to assess BQ overexpression on epirubicin resistance. (**i**) Mice bearing ZR-75 EV tumors were treated with different doses of epirubicin: 0 mg/kg (N = 4), 2 mg/kg (N = 4), and 5 mg/kg (N = 3). (**iii**) Mice bearing ZR-75 BQ overexpressed tumors were treated with different doses of epirubicin: 0 mg/kg (N = 4), 2 mg/kg (N = 4), and 5 mg/kg (N = 4). Representative images of the tumors formed in nude mice treated with or without epirubicin). (**ii**,**iv**) Fold changes of tumor volume were illustrated in growth curves. Data presented as mean ± SD and analyzed by two-way ANOVA (* *p* < 0.05; *** *p* < 0.001; n.s. nonsignificant).

**Figure 5 cancers-12-00533-f005:**
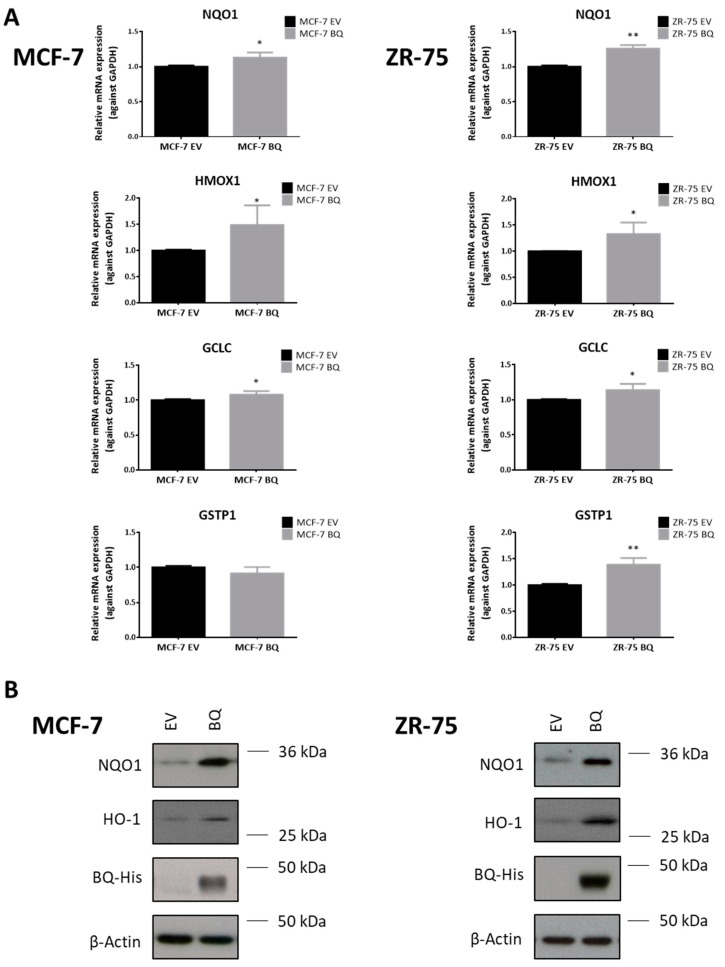
BQ overexpression leads to upregulation of antioxidant enzymes and modulates epirubicin resistance. (**A**) The effect of BQ overexpression on the expression of NQO1, HMOX1, GCLC, and GSTP1 was examined. RT-qPCR was employed to determine mRNA expression level. Data presented as mean ± SD. (Student’s *t*-test, * *p* < 0.05; ** *p* < 0.01; n.s. nonsignificant) (N = 3) (**B**) Western blot was employed to validate the increased expression of NQO1 and HO-1 in BQ overexpressing cells. β-Actin served as loading control (N = 3). (**C**) Expression of NQO1 in primary breast cancer was examined. Immunohistochemical (IHC) staining was employed Representative images of low or high NQO1 expressions were shown. (**D**) Relationship between nuclear BQ expression score and nuclear NQO1 expression score in 124 cases of primary breast cancer samples, demonstrated as scatter plot and linear regression line (R^2^ = 0.4555, *p*-value = 0.0173). (**E**) The effect of either NRF2 or NQO1 downregulation on epirubicin resistance mediated by BQ overexpression. Knockdown of NRF2 or NQO1 were achieved by transfecting specific siRNAs against NRF2 or NQO1 (siNRF2 or siNQO1; 150 pmol) in BQ overexpressing cells. Non-targeting siRNAs were used as control (NSC). At post-transfection 24 h, these cells were seeded in 6-well plates and treated with epirubicin. Clonogenic assay was used to analyze the cell viability of these siRNAs transfected cells under treatment of epirubicin for 14 days. Data presented as mean ± SD. (Two-way ANOVA: * *p* < 0.05; *** *p*  < 0.001) (N = 3).

**Figure 6 cancers-12-00533-f006:**
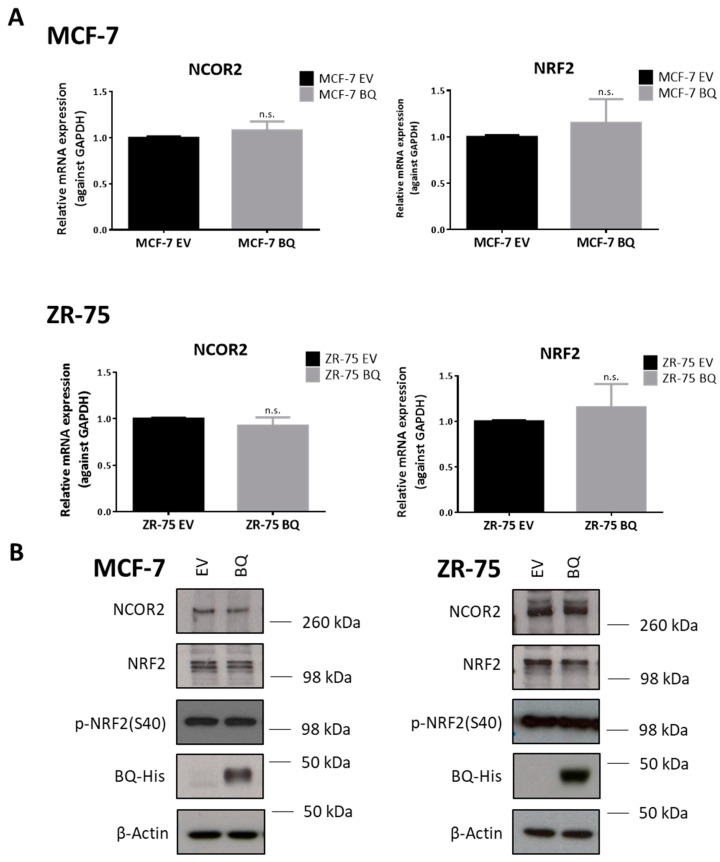
BQ overexpression does not affect the expression or post-translational modification of NRF2. (**A**) Overexpression of BQ did not alter the mRNA levels of both NCOR2 and NRF2. RT-qPCR was employed to determine the expression level. Data presented as mean ± SD. (Student’s *t*-test, n.s. nonsignificant) (N = 3) (**B**) Protein expression of NCOR2, NRF2, phospho-NRF2 (S40) in BQ overexpressing cells was verified by Western blot. β-Actin served as loading control (N = 3).

**Figure 7 cancers-12-00533-f007:**
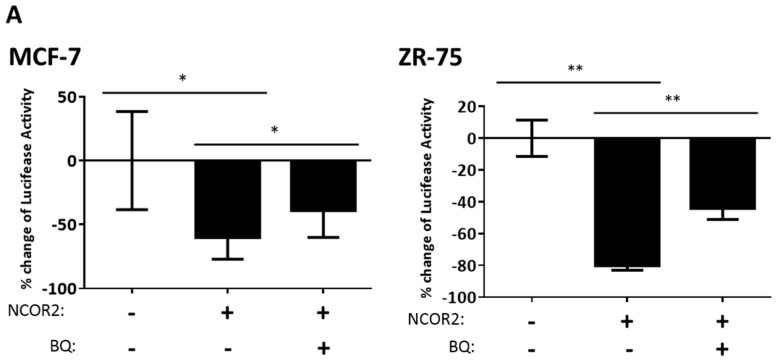
NCOR2 suppresses transcriptional activity of antioxidant response element (ARE) while BQ modulates this effect. (**A**) The effect of NCOR2 and BQ on the activity of ARE was examined by luciferase assay. Luminescence readings were normalized to renilla luciferase activity (as transfection efficiency control). The data were presented as % change of luciferase activity with reference to empty vectors control cells (i.e., BQ and NCOR2 negative). Bars showed as mean ± SD. (Student’s *t*-test, * *p* < 0.05; ** *p* < 0.01) (N = 3) (**B**) The effect of BQ overexpression on the protein interaction between NRF2 and NCOR2. Co-immunoprecipitation was employed to determine the protein–protein interaction. BQ overexpressing cells and control cells were harvested for immunoprecipitation by anti-NCOR2 antibody (IP: NCOR2). Whole cell lysates were used as input. Normal rabbit Immunoglobulin (IgG) was used as pull-down control. (**i**) Immunoblotting by anti-NCOR2, anti-NRF2 and anti-His antibodies were shown. Corresponding bands were indicated by arrows. The intensities of the NRF2 band were quantified by ImageJ software. (**ii**) Bars shown relative bindings of NCOR2 to NRF2 at protein level, presented as mean ± SD. (Student’s *t*-test, * *p* < 0.05; *** *p* < 0.001). (N = 3).

**Figure 8 cancers-12-00533-f008:**
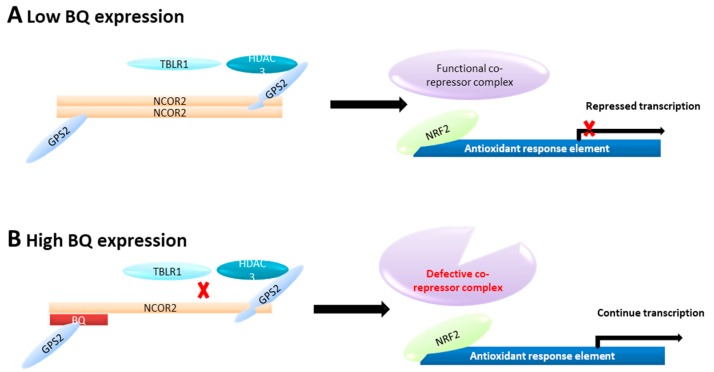
Proposed mechanism of how BQ may contribute to chemoresistance via the regulation of NRF2 in breast cancer. (**A**) Without BQ expression, NCOR2 self-dimerizes in antiparallel fashion, forming a central platform to recruit other corepressor components, such as GPS2, TBL1, and HDACs, to form a functional NCOR2 corepressor complex. The NCOR2 corepressor complex interacts with NRF2 to suppress the transcriptional activity of ARE which is used to regulate the expression of antioxidant genes. (**B**) In the presence of BQ nuclear overexpression, BQ binds with NCOR2This interaction compromises the ability of NCOR2 to fully recruit other corepressor components such as HDACs. This gives rise to a non-fully functional NCOR2 corepressor complex, thus relieving the suppressive effect on NRF2. Upregulation of ARE mediated transcription of antioxidant genes is resulted.

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
