# Peer review of "A Splice Variant of NCOR2, BQ323636.1, Confers Chemoresistance in Breast Cancer by Altering the Activity of NRF2"

_cancers, 2020, doi:10.3390/cancers12030533_

Round 1

Reviewer 1 Report

In this manuscript the authors attempt to establish that the nuclear co-repressor NCOR2 represses NRF2-mediated antioxidant gene transcription and that overexpression of BQ323636, a splicing variant of NCOR2, relieves the repression by forming a heterodimer with and sequestering NCOR2, thereby promoting antioxidant gene expression and resistance to ROS-inducing chemotherapy (eprubicin). Overall, the data presented appear to point to the right direction; however, there are a number of significant deficiencies that make the case less than convincing.

Specific points:

Figures 1 and 2 – The authors tried to establish that epirubicin-resistant MCF-7 cells are also resistant to oxidative stress induced by tBHP and that NRF2 modulates epirubicin resistance in the cells. However, only a single clone of the resistant cells and a single NRF2 shRNA were used. Figure 1 – ROS levels are presented in a bar graph in which the difference is more than 10-fold, which raises the question how the FACS measurement was gated. The original FACS output should be shown. Figure 3 – Why NCOR2 expression was substantially lower in MCF-7 EpiR cells? This should be explained. Figure 4 – BQ expression levels in control and MCF-7 and ZR75 stable overexpression cells should be shown so that the extent of overexpression can be taken into consideration. Figure 5 – The increase of antioxidant gene expression in BQ overexpression cells are modest at mRNA level and not shown at the protein level. Figure 6 – The expression level of BQ should also be measured using an antibody that detect the endogenous protein so that the extent of overexpression can be appreciated. Additionally, what is the percentage of cells that actually overexpress BQ? Figure 7 – The authors tried to establish that NCOR2 suppresses NRF2 activity at the ARE while BQ antagonizes this effect. The results showing the NCOR2-NRF2 coIP are weak and ambiguous as it is unclear if the NRF2 signals/bands were specific. Also, it would be reasonable to expect the authors to show that NCOR2 binds to AREs in the promoters of NRF2 downstream genes by ChIP and that overexpression of BQ disrupt the binding. Overall, NRF2 banding patterns are inconsistent among the figures. Some figures show a typical double band (which is likely correct) but in some figures it is shown as a single band even though the same cells were used, raising the question of antibody specificity or whether the experimental conditions were the same.

Reviewer 2 Report

The article aims to investigate the effects of BQ323636.1, a splice variant of NCOR2, on chemoresistance in breast cancer, and explore whether Nrf2 is involved in the process of resistance. Nrf2 is the main regulator of antioxidant enzyme production and can trigger chemoresistance. This topic is interesting and valuable. However, there are issues need to be explained which listed below.

Major issues:

In the main text, authors describe that BQ interfered the binding of NCOR2 and Nrf2, causing the release and the activation of Nrf2, leading to the production of antioxidant enzymes (NQO1 and HO-1) and the increase of chemoresistance. However, in the reference 21, authors (the same group) mentioned BQ inhibited the binding of ERa and Nrf2, and ERa is the key for tamoxifen resistance by inhibiting Nrf2. Although this article is focused on the epirubicin resistance, please explain the role of ERa in the BQ/NCOR2/Nrf2 regulatory mechanism. In the past, references indicated that once Nrf2 is phosphorylated, it is activated and translocated into nucleus from cytosol, and then bond to ARE for the transcription and translation of antioxidant enzymes. But in figure 2C, the increase of Nrf2 in epirubicin-resistant MCF7 is due to the increase of cytosolic Nrf2, not nuclear Nrf2 (supplementary figure 4). Please explain how cytosolic Nrf2 work? In supplementary figure 2, the luciferase of ARE binding activity is increased in epiR cells, but the nuclear Nrf2 is not increased, please explain how this happen? The detail of the regulation of Nrf2 and the antioxidant enzyme production should be described thoroughly in the introduction section. In figure 2, there should be another experiment about the effect of epirubicin on MCF7 in normal culture condition. In figure 3c, since BQ is in the nucleus (Figure 3C), the authors should observe the expression level of BQ in the nucleus by western blot (Figure 3A). In the Figure 4A, the pictures of colony formation are not correlated to the bar figures. For example, although the colony numbers are higher in the MCF7-BQ under 10 and 20 epirubucin, the colony number is higher in MCF7-BQ compared to MCF7-EV. The MCF7-BQ/MCF7-EC ratio is almost the same in 0, 10, and 20 epirubicin treatment groups. There is no 50% more colony in MCF7-BQ compared to MCF-EV in 10uM epirubicin treatment as shown in the bar figure. The same issue is found in all the colony formation assay. In addition, the statistical comparison of colony formation in Figure 4A should be shown as those in Figure 4 B and C. The significance (*) should be labeled between bars but not the annotations.(the same in Figure 5E) While NAC is efficient in reducing the ROS level in epirubicin+NAC group compared to the epirubicin group in ZR-75, why NAC has no function in MCF7 (epirubicin+NAC group vs epirubicin group)? Since the authors show the same results in both ZR75 and MCF7, why authors choose ZR-75 for animal experiment? In the supplementary Figure 8, in the western blot of BQ overexpression, why the level of BQ is so different in MCF7 in the two BQ overexpression groups? In Figure 5A, can the authors explain why GSTP1 levels are the same in MCF7-EV and MCF7-BQ cells? In Figure 5D, the correlation of BQ and NQO1 should be presented as the correlation figure shown below.

Since ROS is the main focus of this manuscript, the ROS level should be measured in every set of experiments, such as epirubucin treatment, BQ overexpression, siNrf2, siNQO1. In this article, authors show that BQ interferes the binding of NCOR2 to Nrf2 in the epirubicin resistant cells. But, in BQ overexpression cells, there are no changes in the NCOR2 and Nrf2 RNA and protein levels (Figure 2 and 3). It is understandable that the levels of NCOR2 and Nrf2 RNA and proteins don’t change, and it may due to the binding affinity change of NCOR2 and the activity change of Nrf2. However, the p-Nrf2 is not changed in both MCF-7 and ZR-75 with BQ overexpression. The phosphorylation is the key for Nrf2 activity. The author should perform additional experiments for measuring Nrf2 activity throughout the whole manuscript to see if Nrf2 activity is the key for BQ-triggered chemoresistance. In the BQ overexpression cells, the NQO1 and HO-1 are increased, but Nrf2 is not increased. Authors should also explain how NQO1 and HO-1 increased without the activation of Nrf2. The authors should perform additional experiments to see whether siBQ can reverse the epirubicin resistance in MCF7 and ZR-75. In addition, NQO1 overexpression should be performed in EpiR cells with siNrf2 to see if NOQ1 is the most important factor for BQ-triggered epirubicin resistance. The title should be changed to focus on NQO1 instead of Nrf2 according to all the results provided.

Minor issues:

Line 47, “oxidative stress” should be changed to “antioxidant ability”. The effect of tBHP should be described in the result section. In figure 1C, the “relative cell viability” should be changed to “relative colony formation”.

Reviewer 3 Report

The MS by Leung and colleagues reports a novel mechanism of breast cancer chemoresistance which rely on the activation of the transcription factor NRF2 by a splice variant (BQ323636.1) of NCOR2 corepressor. In particular, authors observe an increase in the BQ323636.1 splice variant (BQ) expression in epirubicin-resistant breast cancer cells which also correlates with increased capacity against oxidative stress. They further demonstrate that the splice variant BQ is able to induce chemoresistance in breast cancer by reducing the suppressive function of NCOR2 on NRF2. Indeed ectopic expression of BQ competes with the wt form of NCOR2 for the binding to NRF2 thereby preventing NRF2 inactivation.  Importantly, they also observe that high expression levels of BQ correlates with breast cancer recurrence after chemotherapy and suggest that BQ expression could act as a potential predictor of chemoresistance in breast cancer patients.

Overall the results are of interest but there are several points that must be addressed to strengthen the conclusions.

- In order to better support the evidence that epirubicin resistance correlates to BQ expression, BQ levels must be analyzed in other syngenic models of chemotherapy-sensitive and -resistant cells. In addition, the quality of the figure 3A is poor, higher quality of the WB or different methods for BQ quantification is required.

- Considering the observed increased in BQ expression in MCF7 epi-R cells: is NRF2 binding to NCOR2 compromised in these cells? Is NCOR2 overexpression able to restore sensitivity to epirubicin?

- Are HO-1 levels increased in MCF7 epi-R cells? Does its knock-down confer sensitization to epirubicin?

- In figure 2 and 3, authors show that epirubicin resistant cells display higher NRF2 expression levels as compared to sensitive cells. Furthermore they also observe increased BQ levels in the same cells. What is not clear to this reviewer is the mechanism by which BQ repressor variant could lead to an increase of NRF2 transcriptional activity. Is there and increase co-occupancy of NRF2 targets un epi-resistant cells? Or altered chromatin structure?  

- authors demonstrate that BQ overexpression does not affect the expression, the phosphorylation or the subcellular localization of NRF2, but it does affect its transcriptional activity. Following the point above, is the recruitment of NRF2 at ARE elements affected by BQ expression?

- In the proposed mechanism (Figure 8), BQ binding to NCOR2 results in a non-functional corepressor complex on NRF2 at ARE elements. The authors demonstrate the existence of an interaction between BQ and NRF2, I guess this occurs in the nucleus ?

Moreover:

- It is not clear the meaning of means and SD referring to the quantification of BQ/NCOR2 in figure 3A. Is it represented the mean of protein expression in 3 different clones of sensitive/resistant cells or the same analysis has been repeated 3 times on the same cells?

- In fig 5A GCLC and GSTP1 upregulation upon BQ overexpression is not really convincing.

- It is not clear the number of experimental replicates performed overall in the MS.

- Regarding Fig. 1C, 2B, 4A, and 5E authors should refer to clonogenic ability of cells.

- about the evidences of oncogene induced activation of NRF2, beside the cited article (De Nicola et al, Nature 2011), it is worth mentioning other evidences  which rely on oncogenic KRAS (Tao et al., Cancer Res 2014) and on oncogenic missense mutant p53 protein (Walerych et al., Nat Cell Biol 2016; Lisek et al., Oncotarget 2018).

Round 2

Reviewer 1 Report

The manuscript has been slightly improved. However, the authored failed to address most of the key concerns raised, there are still numerous obvious questions that remain unanswered, and many parts of the data are still not convincing. Therefore, another round of major revision will be necessary before the manuscript can be considered for publication. In addition, it should also be pointed out that the ordering, organization, sizing and labeling of the figures all leave much to be desired.

Reviewer 2 Report

Authors perform experiments and answer all the issues. Now the manuscript is better to understand and more solid.

Author Response

No response was required by this reviewer

Reviewer 3 Report

The authors have addressed some of my concerns but others have not. For example, regarding the effect of NCOR2 overexpression in restoring sensitivity to epirubicin, they comment that required additional experiments go beyond the focus of this study. However, this reviewer believes that understanding the effect of NCOR2 overexpression in the acquisition of chemoresistance in breast cancer is the actual focus of the study. 

Author Response

The reviewer has suggested we demonstrate the effect of NCOR2 overexpression in restoring sensitivity to epirubicin. To address the reviewer’s interest, we have tried to transfect the NCOR2 expression plasmid in to MCF-7 EpiR cells. MCF-7 EpiR cells become more difficult to transfect than MCF-7 cells. Compounded by the large size of the NCOR2 plasmid, our preliminary attempts have not yet yielded success, indicating this experiment will still require much more time to complete. In the last response, we quoted a previous publication [1] investigating its effects in another cancer. Their results showed NCOR2 overexpression in certain HNSCC cell lines would reduce resistance to chemotherapy (cisplatin and paclitaxel) while enhanced resistance was found in other HNSCC cell lines. We agree that adding data of NCOR2 overexpression can be an interesting point, but waiting for these experiments to be completed will result in significant delay to the manuscript publication.

Reference:

Rigalli, J.P.; Reichel, M.; Reuter, T.; Tocchetti, G.N.; Dyckhoff, G.; Herold-Mende, C.; Theile, D.; Weiss, J. The pregnane X receptor (PXR) and the nuclear receptor corepressor 2 (NCoR2) modulate cell growth in head and neck squamous cell carcinoma. PLoS One 2018, 13, e0193242-e0193242, doi:10.1371/journal.pone.0193242.

Round 3

Reviewer 1 Report

Authors have made efforts to improve the manuscript.

Reviewer 3 Report

may accept despite not fully addressed the point